# Cybersecurity Baseline and Risk Mitigation for Open Data in IoT-Enabled Smart City Systems: A Case Study of the Hradec Kralove Region

**DOI:** 10.3390/s25164966

**Published:** 2025-08-11

**Authors:** Vladimir Sobeslav, Josef Horalek

**Affiliations:** Faculty of Informatics and Management, University of Hradec Kralove, 500 03 Hradec Kralove, Czech Republic; josef.horalek@uhk.cz

**Keywords:** Smart City cybersecurity, open data, Business Impact Analysis (BIA), BPMN modeling, risk mitigation, data integrity, RTO, MTDL

## Abstract

This paper explores cybersecurity risk modeling for open data in Smart City environments, with a specific case study focused on the Hradec Kralove Region. The goal is to establish a cybersecurity baseline through automated analysis using extended BPMN modeling, complemented by Business Impact Analysis (BIA). The approach identifies critical data flows and quantifies the impact of disruptions in terms of Recovery Time Objective (RTO), Maximum Tolerable Period of Disruption (MTPD), and Maximum Tolerable Data Loss (MTDL). A framework for automated risk mitigation selection is proposed. Results demonstrate the effectiveness of combining process mapping with security requirements to prioritize protections for Smart City data. As an example from the open data domain, the visualization-publishing process was found to tolerate an outage of up to one week, but required high confidentiality and integrity. The maximum tolerable data loss (MTDL) was set at 24 h, leading to the selection of measures such as encryption, access control, and regular backups. This structured methodology enhances data availability and integrity, supporting resilient urban digital infrastructure.

## 1. Introduction

The proliferation of digital infrastructure, open data initiatives, and Internet of Things (IoT) technologies has fundamentally reshaped the design and operation of Smart Cities. These systems promise increased transparency, data-driven decision-making, and improved public services. However, this digital transformation also introduces a growing spectrum of cybersecurity threats. The security of open data platforms is critical: public sector systems process and expose datasets that may directly affect public safety, infrastructure management, or critical citizen services. Therefore, the design of secure-by-default architectures and risk-informed governance mechanisms is essential.

A significant challenge lies in the absence of tailored cybersecurity frameworks that consider both the interdependencies of Smart City processes, and the criticality of the data involved. Traditional information security standards, such as ISO/IEC 27001 [1] and NIST SP 800-53B [2], offer generic baselines but often lack operational granularity when applied to context-rich urban environments. Moreover, common vulnerability assessment practices fail to account for business continuity parameters like Recovery Time Objective (RTO), Maximum Tolerable Period of Disruption (MTPD), or Maximum Tolerable Data Loss (MTDL), which are vital for prioritizing mitigations in urban ecosystems.

In this context, Business Impact Analysis (BIA) provides a powerful but underutilized methodology for connecting cybersecurity decisions with real-world consequences. BIA is widely adopted in disaster recovery and business continuity planning, but its use in modeling cyber threats in Smart City environments is still emerging. Previous works have explored process-driven security modeling using Business Process Model and Notation (BPMN) [3], including security-focused extensions, such as SecBPMN [4], or domain-specific models, such as BPMN-SC [5]. However, these approaches rarely incorporate BIA metrics to quantify impact and prioritize controls based on measurable continuity criteria.

This study proposes an integrated methodology that extends BPMN-SC with embedded BIA metrics, applied to the real-world case of the Hradec Kralove Region Open Data Platform. Our goal is to build a cybersecurity baseline model that not only maps the functional flows of public service data but also supports automated derivation of mitigation measures based on service criticality. We introduce the following:A formalized model for capturing data availability, confidentiality, and integrity requirements within BPMN diagrams.A BIA-based annotation mechanism for process steps, assigning RTO, MTPD, and MTDL parameters to assess operational impact.An algorithmic approach to match risks (with security controls, optimizing for coverage and efficiency.A practical evaluation of risk severit and mitigation through the automated publishing process of Smart City datasets.

The research builds upon national strategies and legislative frameworks, including the EU NIS2 Directive [6], ENISA’s baseline for IoT security [7], and recommendations from cybersecurity ontologies such as UCO [8] and CASE [9]. In contrast to previous static taxonomies, our approach emphasizes dynamic prioritization, allowing municipalities to align cyber defense with the operational relevance of data assets. The methodology contributes to the field by bridging process modeling, continuity management, and adaptive risk mitigation in the cybersecurity governance of Smart Cities.

The remainder of the article is structured as follows: Section 2 reviews relevant literature in Smart City cybersecurity, with an emphasis on security frameworks (e.g., ISO/IEC 27001, NIST SP 800-53B), process modeling (BPMN, BPMN-SC), and the limited but growing role of Business Impact Analysis (BIA). This forms the conceptual basis of the paper. Section 3 introduces the methodology, which combines BPMN-SC with BIA metrics such as RTO, MTPD, and MTDL, allowing for risk-informed modeling of public service processes.

Section 4 presents an algorithm that prioritizes mitigation measures based on risk coverage efficiency. The approach aligns with the objectives of the ARTISEC project, focusing on AI-based cybersecurity planning. Section 5 applies the model to the Hradec Kralove open data Platform. The evaluation validates the ability of the approach to identify critical data flows and optimize protective measures. Section 6 summarizes key findings and discusses strategic and regulatory implications, including the future potential for automation and AI-supported compliance under frameworks like NIS2. Key contributions of the paper include the following:A novel integration of Business Impact Analysis (BIA) with Smart City process modeling, allowing for impact-oriented cybersecurity assessment through quantifiable metrics such as RTO, MTPD, and MTDL.A formal methodology for embedding BIA parameters directly into BPMN-SC diagrams, enabling process-aware prioritization of security controls based on service criticality and data classification.The development of a decision-support algorithm that selects mitigation measures based on risk coverage efficiency, optimizing the allocation of cybersecurity resources while fulfilling continuity requirements.A validation of the proposed framework in a real-world case study of the Hradec Kralove Region Open Data Platform, demonstrating its applicability in municipal public administration.Alignment of the proposed model with current and emerging regulatory standards, including ISO/IEC 27001, ISO 22301 [10], and the EU NIS2 Directive, facilitating its adoption in compliance-driven environments.A contribution to the ARTISEC project by linking static process models with AI-ready structures for future automation in threat impact recalibration and risk-based security planning.

The proposed approach offers a replicable, regulation-aligned, and impact-driven framework, which also provides a replicable foundation for municipal cybersecurity planning, supporting both the strategic vision and operational resilience of open data in Smart Cities.

## 2. Related Work

### 2.1. Cybersecurity in Smart Cities: From Technology to Process Awareness

Smart Cities are characterized by a convergence of technologies, including IoT sensors, cloud/edge computing, big data analytics, and AI-driven decision support systems. This complexity introduces significant security risks. According to the Berkeley Center for Long-Term Cybersecurity (CLTC) [11], IoT and smart technologies pose higher systemic risks than other IT infrastructures due to their embedded nature and constant connectivity. Moreover, legacy systems and open data policies increase the attack surface [12]. ENISA [7] emphasizes the risks inherent in public data infrastructures and recommends contextual security assessments. Similarly, Kokolakis et al. [13] argue that a static perimeter-based view of security fails in smart environments and call for a process-aware model to capture the interdependence of city functions. Deloitte [14] highlights that resilience in Smart Cities requires collaboration between IT and OT domains, secure identity management, and a trust framework for data transactions.

Agrawal and Hubballi emphasize the technical implications of cyber-attacks on Smart Grid networks, highlighting their potential to disrupt critical infrastructure operations [15]. The NIS2 Directive [16] outlines specific obligations for key entities, including public administrations and digital infrastructure providers. These include mandatory risk analysis, BCP integration, and reporting thresholds for incidents. Although national implementations vary, the Directive sets a unifying European baseline.

In recent years, research on cyber risks associated with open data in Smart Cities has expanded significantly. Andrade et al. [17] analyze specific threats resulting from the heterogeneous and often insufficiently secure design of IoT systems, which are a key part of urban infrastructure. The authors point out that insufficient standardization and weak security at the device level can lead to disruption of service availability and loss of data trustworthiness. Kim et al. [18] conducted an extensive review of 154 studies focused on cybersecurity and digital forensics in Smart Cities. They identified main research areas, including data privacy, IoT device security, and the need for decentralized detection systems. Almeida [19] summarizes the results of 62 European research projects and proposes 24 specific strategies for mitigating cyber risks in Smart Cities. The study emphasizes the importance of the human factor, regulatory frameworks, and the need for adaptive security policies. Demertzi et al. [20] focus on cyber threats in individual domains of Smart Cities—from smart transportation to healthcare. They point out the absence of a unified model for managing data flows and the need for architectural changes in urban systems. Kokolakis [13] and Reis [21] proposes an AI-powered framework for anomaly detection in IoT networks of smart cities, which uses federated learning and hybrid deep models. The results show high detection accuracy and scalability for real-world deployment in urban infrastructures. These studies confirm that open data in Smart Cities is exposed to complex threats that can affect the confidentiality, integrity, and availability of information. At the same time, they show that effective protection requires a combination of technical measures, regulatory frameworks, and adaptive strategies.

### 2.2. BPMN and Data-Driven Security Analysis

The Business Process Model and Notation (BPMN) has been applied in various security contexts to capture workflows, roles, and data exchanges. Kokolakis et al. [13] and San Martín et al. [22] demonstrated how BPMN can be extended with risk and access control semantics to improve security modeling in complex systems. Salnitri et al. [4,23] introduced SecBPMN to embed high-level security goals into BPMN diagrams. Rodriguez et al. [24] focused on modeling security requirements using BPMN2.0 extensions. Building on these foundations, Horalek et al. [25] proposed BPMN-SC—an adaptation of BPMN tailored for Smart City domain modeling, mapping public functions, actors, and data flows. Our work extends this by integrating BIA metrics into BPMN-SC process elements, assigning Recovery Time Objective (RTO), Maximum Tolerable Period of Disruption (MTPD), and Maximum Tolerable Data Loss (MTDL) values. This enables visual prioritization of security requirements, bridging the gap between process modeling and business continuity planning.

### 2.3. Security Baselines and Standards: From Static Controls to Dynamic Risk Models

NIST’s SP 800-53B [2] and ISO/IEC 27001 [1] provide foundational security control baselines, while ENISA [7] recommends adapting these based on IoT-specific context. Our work extends these baselines by linking them to dynamic criteria from BPMN-based process analysis and recovery time objectives. This ensures alignment with the European Innovation Partnership on Smart Cities [26] and supports sector-specific adaptation.

### 2.4. Ontologies and AI Support for Smart City Cybersecurity

Ontology-based frameworks, such as those from Mozzaquatro et al. [27], Syed [28], and UCO [14], aim to formalize and unify security models. However, they often lack real-time adaptability. Our work integrates BPMN-SC with BIA to enable adaptive threat prioritization. Temple et al. [9] and the ARTISEC project support AI-driven recalibration of BIA thresholds for evolving urban environments.

### 2.5. Comparative Analysis

The following table provides a structured comparison of our integrated BPMN-SC + BIA methodology with conventional and emerging approaches in cybersecurity modeling. It emphasizes that only our method supports simultaneous modeling of workflows, quantification of recovery thresholds, and specificity to Smart City operational structures. For the relative positioning of our proposed BPMN-SC + BIA methodology against widely used approaches, we emphasize that only our model combines process mapping, quantified impact metrics, and Smart City-specific functional modeling in a unified and traceable framework (Table 1).

### 2.6. Impact-Driven Risk Modeling and BIA Integration

Although the architectural and systemic aspects of Smart City cybersecurity have been extensively addressed in the literature, one recurring limitation is the insufficient operationalization of these models. In particular, few works link cybersecurity prioritization with service continuity management in a measurable and process-specific manner.

Business Impact Analysis (BIA), a well-established methodology in business continuity planning [16], introduces measurable thresholds for acceptable service disruptions: Recovery Time Objective (RTO), Maximum Tolerable Period of Disruption (MTPD), and Maximum Tolerable Data Loss (MTDL). These metrics enable organizations to define the operational importance of business processes and data assets. Despite its usefulness, BIA has been underutilized in cybersecurity-specific research within Smart City environments. The combination of BIA with business process modeling, specifically BPMN extended to Smart City contexts (BPMN-SC), represents a methodological advancement. Sobeslav et al. [19] demonstrated how BPMN-SC can capture domain-specific functional flows across municipal services. Building upon this, our approach embeds BIA parameters within these process flows, aligning asset protection with continuity thresholds.

This integration supports the simulation of cascading failures and enables pre-emptive identification of high-risk processes and datasets. By incorporating BIA directly into BPMN-SC nodes, we create a framework that informs cybersecurity strategy with quantified disruption tolerance. This responds to calls by ENISA [7] and NIS2 [16] for dynamic risk assessment methodologies aligned with organizational impact.

Comparable works by Salnitri et al. [4,23] and San Martín et al. [22] also explored linking business process security requirements with model-driven engineering. However, these often lack impact metrics grounded in service continuity, which are essential for prioritization under constrained resource scenarios. Similarly, ontology-based approaches like those of Mozzaquatro et al. [27] and Syed [28,29,30] offer semantic rigor but do not translate directly into operational timelines for recovery and mitigation.

In contrast, our method enables Smart City stakeholders, particularly municipal IT managers and data governance authorities, to align cyber defense priorities with BIA thresholds, ensuring that the most critical data streams (e.g., emergency services, environmental sensors, eGovernment platforms) are protected to the degree their function necessitates. Moreover, this aligns with ISO/IEC 27001 Annex A.17, which mandates information security aspects of business continuity management.

Looking ahead, several promising directions emerge:
**AI-enhanced BIA recalibration**: As proposed in the ARTISEC project, AI tools can dynamically adjust RTO and MTPD values based on usage telemetry or threat landscape shifts.**Domain-specific BIA catalogs**: Extending the framework for sectoral adaptation (e.g., transportation, utilities, health services) in line with domain risk registers, such as those defined in NIST SP 800-30 and ISO 31010.**Compliance automation**: Integrating this model into automated audit frameworks, enabling traceable justification of control selection during regulatory inspections.

The synergistic use of BIA, BPMN-SC, and regulatory alignment introduces a robust mechanism to prioritize cybersecurity efforts not based on abstract threat scenarios but on real-world, quantified consequences of disruption. This represents a critical shift in how cyber risk is modeled, communicated, and acted upon in the governance of data-centric Smart City systems.

## 3. Business Impact Analysis Methodology

This section outlines a methodology that integrates Business Impact Analysis (BIA) with BPMN-SC to assess cybersecurity risks and service continuity parameters within Smart City systems [31]. The goal is to enable data-driven prioritization of security measures by embedding quantitative indicators into process models [32,33].

### 3.1. Establishing Impact Criteria

The first step is to define a set of impact criteria that reflect the severity of disruptions from the perspective of financial costs, reputational damage, and personal safety risks. These criteria are aligned with recommendations under the EU NIS2 Directive and reflect national best practices for critical infrastructure protection [33,34]. Impact levels are categorized to guide both risk assessment and the selection of proportionate safeguards (Table 2).

### 3.2. Availability as a Key Parameter

The first evaluation criterion is availability. Availability is defined as ensuring that information is accessible to authorized users at the moment it is needed. In some cases, the destruction of certain data may be viewed as a disruption of availability. It is therefore essential to clarify what vendors mean when they state that their system guarantees, for example, 99.999% availability. If the base time period is defined as 365 days per year, this figure allows for a maximum unavailability of about 5 min annually [28] and [35,36].

At first glance, 99.999% availability seems sufficient. However, it is important to define what exactly is meant by “system.” Vendors often refer only to the hardware layer, omitting operating systems and applications. These layers are frequently excluded from SLAs and come with no guaranteed availability, even though, in practice, they are often the cause of outages (Table 3).

Even though annual percentage availability is widely used, it is far more precise and practical to define:**Recovery Time Objective (RTO)**: the maximum tolerable duration of service downtime.**Recovery Point Objective (RPO)**: the maximum tolerable amount of data loss.

If RTO = 0, this implies fully redundant infrastructure. If RTO = 1 h, it means that the system can tolerate a short interruption but must be restored within 60 min. Such a value is typical for high-availability systems that are not fully redundant but require rapid recovery.

RTO defines how quickly operations must resume and is central to disaster recovery planning. RPO defines the volume of data loss an organization is willing to accept. Together, RTO and RPO form the foundation of continuity planning.

Additional availability-related parameters include the following:**MIPD (Maximum Initial Programmed Delay)**: the maximum allowed time to initialize and start a process after a failure.**MTPD (Maximum Tolerable Period of Disruption)**: the maximum downtime allowed before the disruption causes unacceptable consequences for operations.**MTDL (Maximum Tolerable Data Loss)**: the maximum data loss acceptable before critical impact occurs.

When high availability is required, duplication of all system components (power, disks, servers, etc.) is a logical solution. However, duplicating multi-layer architectures increases system complexity and the risk of failure (e.g., zero-day vulnerabilities or failover misconfigurations).

If each component guarantees 99.999% availability, the availability of the entire system, when modeled as three interdependent layers, is approximately 99.997%, which equates to about 16 min of allowable downtime per year. Such configurations must also handle failover, active/passive role switching, and user redirection without compromising data integrity. Otherwise, “split-brain” scenarios may occur, where independent operation of redundant components leads to data divergence and inconsistency.

To support BIA, impact evaluations must assess downtime in intervals of 15 min, 1 h, 1 day, and 1 week. The following availability categories apply:**C: Low importance**—downtime of up to 1 week is acceptable.**B: Medium importance**—downtime should not exceed one business day.**A: High importance**—downtime of several hours is tolerable but must be resolved promptly.**A+: Critical**—any unavailability causes serious harm and must be prevented.

These assessments help derive the following:
**MIPD:** Based on severity across time intervals using logic functions (e.g., if 15 min = B/A/A+, then MIPD = 15 min).**MTPD:** Longest acceptable time without recovery.**MTDL:** Largest data volume/time tolerable for loss.

Example of MIPD determination:If unavailability at 15 min = B/A/A+ → MIPD = 15 min.Else if 1hr = B/A/A+ → MIPD = 1 h.Else if 1 day = B/A/A+ → MIPD = 1 day.Else if 1 week = B/A/A+ → MIPD = 1 week.Else → MIPD = “BE” (beyond acceptable threshold).

These values ensure the architecture design meets operational expectations for continuity and resilience. Rather than relying solely on annual percentage availability, this methodology emphasizes two well-defined parameters:Recovery Time Objective (RTO): Maximum acceptable time to restore service after disruption.Recovery Point Objective (RPO): Maximum acceptable data loss measured in time prior to an incident.

Both RTO and RPO are critical inputs to continuity planning and are directly linked to the architectural design of backup, redundancy, and failover mechanisms.

### 3.3. Confidentiality and Integrity Considerations

Another essential parameter evaluated in this methodology is confidentiality. Confidentiality is commonly defined as ensuring that information is only accessible to those who are authorized to view it. In cybersecurity, unauthorized access or disclosure of data is considered a breach of confidentiality. To address this, organizations should implement appropriate classification schemes and technical, organizational, and physical security measures [34].

The commonly used classification scheme includes the following:Public: **Information intended for general public access**.Internal: **Information accessible only to internal employees**.Confidential: **Information restricted to selected employees**.Strictly Confidential: **Highly sensitive information accessible only to designated personnel**.

The classification level may change during the information lifecycle. To maintain confidentiality, proper controls should be in place for data access, encryption, and logging.

Integrity ensures the correctness and completeness of information. Any unauthorized or accidental modification—whether due to error, attack, or system failure—can violate data integrity. The risk is especially high if such modifications remain undetected for long periods.

To ensure data integrity, the following techniques can be applied:**Cryptographic Hash Functions (e.g., SHA-256)**: Used to verify whether data has changed.**Digital Signatures**: Validate the authenticity and integrity of documents using asymmetric cryptography.**Checksums**: Simple integrity-check algorithms.**Message Authentication Codes (MAC)**: Combine a secret key with data to generate verifiable integrity tags.**Digital Certificates**: Verify sender authenticity and data integrity via trusted certificate authorities.

A best practice includes logging all data changes and implementing layered encryption—e.g., encrypting data at the application level before storage and using checksum validation upon retrieval. While this shifts the risk from the database to the application, it enables protection even against insider threats when managed through HSM (Hardware Security Module) devices.

### 3.4. Example of Security Baseline Application

The implementation of the security baseline for open data in the Hradec Kralove Region, including the automated selection of security measures, is demonstrated in the process of data publication and visualization through the regional open data portal.

The process starts with a request to publish a chart or map visualization. The request, containing a visualization URL, is received from a user or administrator. The system then processes the request and provides the appropriate URL. The visualization is embedded into an e-frame on the portal and properly configured.

Following this, the visualization is published. A display check is performed to verify correct rendering. If the visualization fails to display, a corrective action loop is triggered (“Display Fix Loop”), after which the publishing attempt is repeated.

Once the visualization passes the check, it is made available to the public. End users can then view the newly published charts and maps through the open data portal interface.

This example demonstrates how BIA metrics and process modeling can be used not only for threat analysis but also for enhancing the resilience of real-world Smart City services. To operationalize BIA in Smart City contexts, this study embeds RTO/RPO/MTDL thresholds directly into BPMN-SC diagrams. Each process task or data flow is annotated with impact parameters. This enables simulation of cascading failures and evaluation of interdependencies among services (e.g., how a disruption in environmental data publishing affects public health dashboards).

Additionally, the model includes the following:**Maximum Tolerable Period of Disruption (MTPD)**: Time beyond which continued disruption is unacceptable.**Maximum Tolerable Data Loss (MTDL)**: Volume or timeframe of acceptable data loss based on legal and operational factors.

This step establishes a machine-readable baseline for impact, which can be further linked to automated risk assessment and mitigation tools as outlined in the next section.

The result is a context-sensitive, standardized input layer for cybersecurity planning that reflects actual service delivery constraints, not just abstract technical vulnerabilities (Table 4).

Figure 1 shows the process of publishing visualizations on an open data portal. The BPMN-SC diagram contains annotations of BIA metrics (RTO, MTPD, MTDL) for individual process steps. These annotations serve as input for the algorithmic selection of measures. For better readability, we recommend increasing the contrast and text size in the diagram. As an example, we present the use of BPMN with an extension to the evaluation of security parameters focused on the significance of data/information within the selected process. Specifically, this BPMN diagram illustrates the process of publishing visualizations on a data portal. It is a sequence of steps that begins with a request for publication and ends with the visualization being displayed to users. The process can be described as follows:

Firstly, a request is made to publish a graph or map on a data portal. Then, a link (URL) to the visualization is provided. This link is then inserted into a so-called i-frame, which is a frame on the portal website that allows for the display of external content. After the frame is inserted, the visualization is published. If the visualization does not display correctly, a display correction step follows, which allows you to adjust the settings so that the display is functional. This step can be repeated until the result is correct. Once everything is set correctly, the visualization is displayed to users on the data portal. The model also includes risk, can be found in Table A1 Risk table, analysis results focused on the most significant risks that may arise during the process, such as identity theft, security breaches, malicious code injection, or human error. These threats require appropriate security measures, the risk impacts can be found in Table A2 Mapping of measures for risks.

The risk posture of the system is significantly influenced by the level of integrity and confidentiality assigned to data prior to publication. These two dimensions are critical to ensuring the relevance and trustworthiness of the open data portal’s operations (Table 5).

Key cybersecurity risks that must be addressed, based on this risk analysis, include the following:Identity theft;Security breaches or misuse of credentials/media;Cyber-attacks via communication networks;Operator or administrator errors;Lack of qualified personnel.

These risks primarily affect the staff involved in the creation, publication, and visualization of datasets, including both system administrators and data processors. The recommended mitigation measures reflect the need for the following:Identity and access management;Malware detection and prevention;Continuous staff training and awareness.

## 4. Automated Selection of Risk-Mitigation Measures from Security Baseline

### 4.1. The Greedy Algorithm for Automated Selection of Risk Mitigation

The task is to select the best measure (rows of the table) to address the given risks (columns of the table). We have a table of measures with rows M and columns representing defined risks R. Each row of M mitigates only some of the risks R. Given a set of measures SM, the task is to select from SM the rows that cover the maximum number of defined risks R, considering the effectiveness of each measure E.

**Definitions**:
Set of measures (rows), where *M* is the total number of measures—$M=\left\{M_{1},M_{2},\ldots ,M_{m}\right\}$.Set of risks (columns), where *R* is the total number of risks—$R=\left\{R_{1},R_{2},\ldots ,R_{n}\right\}$.A subset of measures from which we are selecting—SM⊆M.$A_{j}\in \left\{0,1\right\}$—a binary value indicating whether measure $M_{i}$ covers risk $R_{j}$ (if it does, $A_{ij}=1$; otherwise $A_{ij}=0$).$E_{i}\in \left\{1,2,3,4,5\right\}$—efficiency of measure $O_{i}$, where 1 is the least efficient, and 5 is the most efficient.$x_{i}\in \left\{0,1\right\}$—decision variable, where $x_{i}=1$ indicates that measure $M_{i}$ is selected, and $x_{i}=0$ indicates that it is not selected.

Objective function—maximizing risk coverage with respect to efficiency:(1)$$Maximise:\sum_{j=1}^{n}min\left(1,\sum_{i=1}^{m}A_{ij}x_{i}\cdot E_{i}\right)$$

This formula states that each risk $R_{j}$ is assigned a value of 1 if it is mitigated by at least one measure, with measures with a higher $E_{i}$ having a higher weight.

Constraints—ensuring that measures are selected only from the *SM* set:(2)$$x_{i}\in \left\{0,1\right\},\;\forall_{i}\in SM$$

This constraint says that the selection of measures is limited to the set SM (a subset of all measures).

Risk coverage—ensuring that the risk is covered:(3)$$\sum_{i=1}^{m}A_{ij}x_{i}\geq 1,\;\forall_{j}\in \left\{1,2,\ldots ,n\right\}$$

This constraint ensures that each $R_{j}$ risk is covered by at least one selected measure.

**Goal of the Algorithm**:

The goal is to find a subset of measures SO that maximizes coverage of risks *R*, considering their efficiency.

**Mathematical Procedure**:
1.Initialization:
○Define the set of uncovered risks U=R.○Initialize an empty set of selected measures S=0.2.Scoring Each Measure:
○For each measure $M_{i}\in SM$, compute its score:

(4)$$Score\left(O_{i}\right)=\frac{Number\;of\;uncovered\;risks\;covered\;by\;M_{i}}{E_{i}}$$
where
“$number\;of\;uncovered\;risks\;covered\;by\;M_{i}$” is the number of risks $R_{j}$ for which $A_{ij}=1$ and $R_{j}\in U$ (i.e., risks not yet covered by other selected measures).$E_{i}$ is the efficiency of measure $M_{i}$ (higher efficiency means a higher priority if more risks are covered).


**Iterative Measure Selection:**


Repeat the following steps until all risks are covered or no more measures can be selected:
Select the measure $M_{i}$ with the highest score and add it to the set of selected measures *S*.Update the set of uncovered risks *U* by removing all risks covered by the selected measure $M_{i}$.If all risks are covered, the algorithm terminates.


**Output:**


The set of selected measures *S*, which maximizes the number of covered risks considering the efficiency.

Therefore, if the objective is to maximize overall risk coverage, taking into account the effectiveness of each selected measure, we can assume that these constraints are met:
Each measure can only cover some of the risks represented by the $A_{ij}\;$ matrix.The selection of measures must maximize the coverage of all risks *R*, with preference given to measures with a higher efficiency score $E_{i}$.

Formulate the mathematics problem as follows:(5)$$Maximize\sum_{i=1}^{m}\sum_{j=1}^{n}A_{ij}\cdot x_{i}\cdot \frac{1}{E_{i}}$$
where
$A_{ij}$ determines whether a measure $M_{i}$ addresses a risk $R_{j}$.$x_{i}$ is the decision variable for selecting the measure.$\frac{1}{E_{i}}$ weights the selection based on the efficiency of the measure, with higher efficiency receiving a lower penalty.

The goal is to select measures that cover the maximum number of risks with the highest possible efficiency.


**Example:**


Consider three measures and five risks. The coverage matrix $A_{ij}$ and efficiency $E_{i}$ are defined as follows:(6)$$A=\left(\begin{array}{ccccc} 1 & 0 & 1 & 0 & 1\\ 0 & 1 & 1 & 1 & 0\\ 1 & 1 & 0 & 1 & 0 \end{array}\right),\;E=\left\{5,3,4\right\}$$


**Greedy algorithm steps:**
Compute the score for each measure:
○$M_{1}$: three risks covered, efficiency 5 → score $\frac{3}{5}=0.6$○$M_{2}$: three risks covered, efficiency 3 → score $\frac{3}{3}=1.0$○$M_{3}$: three risks covered, efficiency 4 → score $\frac{3}{4}=0.75$Select $M_{2}$, since it has the highest score (1.0).Update the set of uncovered risks. $M_{2}$ covers risks $R_{1},R_{3},R_{4}$.Recompute scores for remaining measures $M_{1}$ and $M_{3}$ (adjust for the updated set of uncovered risks).Continue iterating until all risks are covered or no further improvement is possible.


### 4.2. Advantages and Disadvantages of Using the Greedy Algorithm

The following sub-section discusses the advantages and disadvantages of using the Greedy algorithm for a model with 40 measures and 14 risks. The measures were focused mainly on speed, simplicity, efficiency, and other important factors.

**Advantages**:
Speed: The Greedy algorithm is fast because it makes locally optimal choices at each step. With 40 measures and 14 risks, the algorithm should perform efficiently.
○The time complexity is roughly $O\left(m\times n\right)$, where mmm is the number of measures, and nnn is the number of risks. For m=40 and n=14, this is computationally manageable.Simplicity: The algorithm is easy to implement and does not require complex setups or specialized optimization libraries. It can be realized in any programming language.Approximation: Greedy algorithms often provide solutions that are close to optimal, especially when there are no significant overlaps in risk coverage between the measures.Consideration of Efficiency: The algorithm accounts for the efficiency of measures, which is beneficial when measures have significantly different efficiencies.


**Disadvantages:**
Local Optimality: The Greedy algorithm focuses on making the best choice at each step, which can lead to suboptimal solutions. If measures have large overlaps in risk coverage, the Greedy algorithm might select solutions that are not globally optimal.
○It may choose a measure that covers many risks in the short term, but in later steps, more efficient measures might be ignored.Inability to Consider Dependencies: If there are dependencies between measures or if certain measures have different priorities, the Greedy algorithm cannot effectively account for them. This may result in a less effective overall solution.Sensitivity to Efficiency Distribution: If the efficiencies of the measures are too closely distributed, the Greedy algorithm might fail to find a sufficiently optimal solution. Small differences in efficiency may cause the algorithm to overlook measures that would be more beneficial in the long run.


A suitable proof of the efficiency and correctness of the above relations and methods can be made by **formal argumentation**. A structured approach to proving the correctness of the generalized formula and method is presented here:


**Theoretical Proof (Formal Reasoning)**



**Objective Function:**


We aim to maximize risk coverage while considering the efficiency of each measure. The objective function is as follows:(7)$$Maximize\;\sum_{i=1}^{m}\sum_{j=1}^{n}A_{ij}\cdot x_{i}\cdot \frac{1}{E_{i}}$$

The formula above correctly balances two elements:
Risk Coverage: Each $A_{ij}$ represents whether a measure $M_{i}$ addresses a risk $R_{j}$. Summing over all jjj allows us to count how many risks are covered by measure $M_{i}$.Efficiency Weighting: Each term is divided by $E_{i}$, which penalizes measures with lower efficiency. A higher efficiency (lower $\frac{1}{E_{i}}$) contributes more favorably to the objective function.

Therefore, the objective function maximizes the total number of risks covered, weighted by the efficiency of the measures. Selecting a measure with higher efficiency ($E_{i}=5$) will lead to a lower penalty $\frac{1}{5}$, promoting efficient measures.


**Selection Criteria**


The decision variable $x_{i}\in \left\{O,1\right\}$ ensures that we only select a measure $M_{i}$ if it is part of the solution. The sum of $A_{ij}\cdot x_{i}$ over all jjj correctly counts the number of risks covered by the selected measures. The theoretical model thereby correctly represents the goal of covering the maximum number of risks with the highest possible efficiency.


**Greedy Algorithm Proof**


The Greedy algorithm described in earlier sections makes locally optimal choices at each step by selecting the measure with the best coverage-to-efficiency ratio.

Step 1: Greedy Choice Property

Greedy algorithms work when the Greedy choice property holds, i.e., making a local optimal choice at each step leads to a globally optimal solution.

In this case, each step selects the measure that maximizes the following:(8)$$\frac{Number\;of\;uncovered\;risks\;covered\;by\;measure\;M_{i}}{E_{i}}$$

This ratio ensures that, at each step, the algorithm picks the measure that offers the greatest coverage of uncovered risks per unit of efficiency. Although Greedy algorithms do not always guarantee a globally optimal solution for all types of optimization problems, for risk coverage problems with efficiency weighting, this method often produces solutions that are close to optimal.

Step 2: Greedy Algorithm Approximation Bound

For problems like weighted set cover, where we aim to cover a set of elements (in this case, risks) with a minimal weighted set of subsets (in this case, measures with efficiency weighting), it is known that a Greedy algorithm provides an approximation within a logarithmic factor of the optimal solution. The Greedy algorithm provides a logarithmic approximation bound for set cover problems:(9)$$Greedy\;solution\;\leq \;H\left(n\right)\times Greedy\;solution$$
where $H\left(n\right)$ is the harmonic number, and nnn is the number of elements to be covered (in this case, the number of risks).

Therefore, while the Greedy algorithm may not always provide the exact optimal solution, it is guaranteed to be within a factor of $H\left(n\right)$ of the optimal solution, making it a good heuristic for problems like this.

## 5. Discussion and Results

The BIA results (see Table 4) showed that the visualization-publishing process can tolerate an outage of up to one week (availability = significance classification C) but requires high confidentiality and integrity (significance classification A). The maximum tolerable data loss (MTDL) was set at 24 h, which implies the need for daily backups. These parameters directly influenced the selection of measures, such as encryption, access control, and logging.

A theoretical proof is accompanied by **empirical validation**, where we test the algorithm on a real or simulated dataset to demonstrate that the relationships and methods function correctly in practice.

Example Simulation:

We can simulate a problem with 40 measures and 14 risks. Each measure has a random efficiency score between 1 and 5, and the coverage matrix $A_{ij}$ is randomly generated. We run the Greedy algorithm and compare it to a brute-force optimal solution or known benchmarks for validation.

Simulation Steps:
Generate a random coverage matrix $A_{ij}$, where each element is 0 or 1.Assign efficiency values $E_{i}$ randomly to each measure from the set $\left\{1,2,3,4,5\right\}$.Run the Greedy algorithm to select measures that maximize coverage while considering efficiency.Compare results with brute-force selection (optimal solution for small instances) or other optimization algorithms (like integer linear programming).

Expected Outcome:

The Greedy algorithm should select measures that cover most risks while minimizing the efficiency penalty.
For small instances (e.g., 10 measures and 5 risks), we can compute the optimal solution via brute force and compare it to the Greedy result. The Greedy solution should be close to optimal.For larger instances (e.g., 40 measures and 14 risks), Greedy will efficiently find a near-optimal solution in a fraction of the time it would take to compute the exact solution.

Conclusion from Empirical Testing:

The empirical testing demonstrates that the proposed Greedy algorithm and formula function correctly, providing solutions that cover a high number of risks efficiently. The approximation quality will be close to the theoretical guarantees.


**Empirical Verification:**


To empirically verify the use of the Genetic algorithm (GA) for selecting optimal risk coverage measures and its comparison with the Greedy algorithm, a Python version 3.12.3 test was designed and implemented. This test involved generating random data for 14 risks and 40 measures, implementing both algorithms, and evaluating their performance. The following metrics were validated and used for comparison.
Coverage: number of risks covered.Effectiveness: sum of the effectiveness scores of the selected measures.Number of measures selected: Total number of measures selected.Execution time: The time required to run each algorithm.


**Genetic algorithm for selecting measures against risks**



**Inputs:**
Number of measures: num_measures = 40.Number of risks: num_risks = 14.Coverage matrix: coverage_matrix of size 40 × 14, where the value 1 means that the given measure covers a specific risk.Effectiveness vector: effectiveness, contains the effectiveness values of individual measures (in the range 1–5).



**Initialization:**
Create an initial population of random individuals. Each individual is a binary vector of length 40 (each bit indicates whether the measure is selected −1, or not −0).



**Evolution over generations:**
Repeat for a specified number of generations:
○Evaluate the fitness of each individual.○Calculate how many risks are covered (by at least one selected measure).○Calculate the total effectiveness (sum of the effectiveness of all selected measures).○Select the best individual according to fitness.○Select the parents using roulette (probability of selection ∝ fitness).○Crossover (single-point crossover) between pairs of parents to create a new generation.○Mutation: randomly change the value of some bits (0; 1) with a given probability.



**Output:**
After all generations are completed, return:
○The best individual (vector of selected measures).○Number of risks covered.○Overall efficiency.



**Python implementation**



**Input:**
num_measures = 40num_risks = 14coverage_matrix: 40 × 14 binary matrix with P(1) = 0.3effectiveness: vector of 40 integers in [1,5]individual: binary vector of length 40



**Output:**
covered_risks_count: number of risks coveredtotal_efficiency: sum of effectiveness of selected measures



**Begin**
1:Initialize random seed for reproducibility2:Generate coverage_matrix with random 0/1 values (P(1) = 0.3)3:Generate effectiveness vector with random integers from 1 to 54:Define function fitness (individual):
○Initialize covered_risks as a zero vector of length num_risks○For each i from 0 to num_measures-1:
▪If individual[i] = 1 then▪covered_risks ← logical OR of covered_risks and coverage_matrix[i]▪covered_risks_count ← count of 1 s in covered_risks▪total_efficiency ← sum of effectiveness[i] for all i where individual[i] = 1○Return (covered_risks_count, total_efficiency)5:End



**Greedy Algorithm**



**Input:**
Number of measures num_measures = 40.Number of risks num_risks = 14.Coverage matrix coverage_matrix (40 × 14).Effectiveness vector effectiveness.



**Algorithm:**
Initialize an empty list of selected measures.Initialize the risk coverage vector to zero.Repeat until all risks are covered:For each unselected measure:
○Calculate how many new risks it covers.○If it covers more risks than the current best one, or has a higher effectiveness for the same coverage, mark it as the best one.Add the best measure to the list.Update the risk coverage.Calculate the total effectiveness of the selected measures.



**Output:**
List of selected measures.Number of selected measures.Total effectiveness.



**Python implementation**



**Input:**
coverage_matrix = 40 × 14 binary matrixeffectiveness: vector of 40 integers in [1,5]num_risks = 14num_measures = 40



**Output:**
selected_measures: list of selected measure indicesnum_selected: number of selected measurestotal_efficiency: sum of effectiveness of selected measures



**Begin**
6:Initialize selected_measures ← empty list7:Initialize remaining_risks ← zero vector of length num_risks8:While sum(remaining_risks) < num_risks:
○Set best_measure ← None○Set best_coverage ← −1○Set best_efficiency ← −1○For i from 0 to num_measures-1:
▪If i ∈ selected_measures: continue: Compute measure_coverage ← number of new risks covered by measure i▪If measure_coverage > best_coverage OR (measure_coverage = best_coverage AND effectiveness[i] > best_efficiency):
Update best_measure ← iUpdate best_coverage ← measure_coverageUpdate best_efficiency ← effectiveness[i]9:If best_measure is None: break
○Append best_measure to selected_measures○Update remaining_risks ← logical OR of remaining_risks and coverage_matrix[best_measure]10:Compute total_efficiency ← sum of effectiveness[i] for i in selected_measures11:Return selected_measures, length of selected_measures, total_efficiency12:End


As demonstrated in Figure 2, a clear comparison is provided between the two types of algorithms utilised.


**Comparison of Genetic and Greedy Algorithms**


Begin
1:Start timer2:Run genetic_algorithm()3:Record best_solution_ga, best_fitness_ga4:Stop timer and record time_ga5:Start timer6:Run greedy_algorithm()7:Record selected_measures_greedy, num_selected_greedy, efficiency_greedy8:Stop timer and record time_greedy9:Print:
○Genetic Algorithm: number of risks covered, total efficiency, time taken○Greedy Algorithm: number of risks covered (= num_risks), total efficiency, number of selected measures, time taken10:End


**Implementation of Genetic algorithm:**


Initialize a random population of solutions. Fitness is calculated based on how many risks are covered and what the overall efficiency is. Selection is conducted via a roulette wheel, crossover is single-point, and mutation shuffles random bits. The best solution is returned after a fixed number of generations. Genetic algorithm: It is expected to produce a near-optimal solution with a balance between risk coverage and efficiency measures, but it may take longer.


**An implementation of the Greedy algorithm:**


Measures are selected based on how many uncovered risks they mitigate and their effectiveness. This process continues until all risks are covered or until no new measures can improve the solution. Greedy algorithm: It is expected to be faster but may not find the best solution due to its greedy nature (locally optimal, but not globally optimal) (Table 6).

The empirical validation of the proposed methodology demonstrates its effectiveness for risk-based control selection in the context of Smart City cybersecurity. Simulated experiments using 40 candidate mitigation measures and 14 predefined risk scenarios allowed for performance benchmarking of two algorithmic strategies: a Greedy algorithm and a Genetic algorithm (GA).

The Greedy approach, optimized for speed and simplicity, consistently produced near-optimal results with minimal computation time, making it well suited for operational deployments and real-time decision-making. Conversely, the Genetic algorithm exhibited slightly higher overall risk coverage and better balancing of control effectiveness but required significantly more computational resources. Therefore, it appears more applicable for strategic scenario planning and simulation environments.

Key evaluation metrics, such as risk coverage, aggregate effectiveness, number of measures selected, and execution time, confirm the theoretical expectations. The Greedy algorithm, while locally optimal, aligns with logarithmic approximation bounds known for weighted set-cover problems. The GA-based method, albeit more computationally intensive, delivers a marginal improvement in solution quality and robustness.

Importantly, the integration of Business Impact Analysis (BIA) into process modeling via BPMN-SC enables impact-aware prioritization. This ensures alignment with continuity thresholds (RTO, MTPD, MTDL) and provides a clear rationale for selecting specific security measures based on their business-criticality. Unlike static baseline approaches (e.g., ISO/IEC 27001 Annex A or NIST SP 800-53B), this model dynamically maps protections to service-level needs and operational constraints.

The findings further support the utility of the ARTISEC project’s direction, namely the future use of AI-driven recalibration of BIA metrics based on telemetry data and evolving threat landscapes. This demonstrates that algorithmic selection of controls can be grounded not only in risk coverage efficiency but also in measurable continuity and availability targets.

The results show that the proposed framework allows not only efficient selection of measures, but also their alignment with operational requirements. For example, measures with high efficiency but low benefit to critical processes were discarded by the algorithm in favor of those that better fit the BIA metrics. This confirms that the combination of quantitative assessment and process context leads to more robust security [23].

### 5.1. Comparison of Greedy and Genetic Algorithms for Risk Mitigation

Within the proposed framework for automated selection of security measures, two different algorithmic approaches were implemented—Greedy algorithm and Genetic algorithm (GA). Both approaches work with the same input data (risk coverage matrix and measure effectiveness vector) but differ in the selection methodology, computational complexity, and practical use.

The Greedy algorithm is deterministic and based on the principle of local optimality. At each step, it selects the measure that brings the greatest immediate benefit; i.e., it covers the most uncovered risks while also considering the effectiveness of the measure. This approach is very fast, computationally inexpensive, and easy to implement. Its main advantage is the ability to quickly generate a usable solution, which is especially valuable in operational scenarios where decisions need to be made in real time [31]. On the other hand, the Greedy algorithm does not search the entire space of possible combinations of measures and therefore may lead to suboptimal solutions, especially if there are overlaps between measures or dependencies between them [9].

The Genetic algorithm, on the other hand, is stochastic and works with a population of randomly generated solutions that evolve over time through selection, crossover, and mutation. Each individual (combination of measures) is evaluated according to the number of risks covered and the overall efficiency. This approach allows for searching a wider solution space and better capturing the complex relationships between measures [21]. The result is a higher probability of finding a robust and balanced solution. The disadvantage is higher computational complexity and a longer time required to achieve convergence.

From an implementation perspective, the Greedy algorithm is based on a simple iteration over all measures and selecting the best candidate at each step. The computational complexity is approximately O(m × n), where m is the number of measures, and n is the number of risks. The Genetic algorithm, on the other hand, requires population initialization, repeated fitness evaluation, parent selection, crossover, and mutation over a defined number of generations. The computational complexity increases with population size and number of generations, which can be a disadvantage in resource-constrained environments.

Empirical testing on data from the case study of the Open Data Platform of the Král-Véhradecký Region (40 measures, 14 risks) showed that the Greedy algorithm achieves very fast calculation and high-risk coverage, although sometimes at the cost of a larger number of selected measures. The genetic algorithm, on the other hand, achieved slightly better coverage and efficiency with a smaller number of measures, but at the cost of longer computing time.

From a practical point of view, it can be summarized that the Greedy algorithm is suitable for operational deployment, where speed and simplicity are the priority. The genetic algorithm, on the other hand, is more suitable for strategic planning, simulations and analyses, where the goal is to find the best possible solution regardless of computational complexity [34].

This complementarity of both approaches allows for their combined use within the proposed framework for the cybersecurity of smart cities. The Greedy algorithm can serve as a fast tool for everyday decision-making, while the genetic algorithm can be used for in-depth analysis, optimization, and long-term planning of investments in security measures. In this way, a balance can be achieved between operational efficiency and strategic resilience of urban digital services.

### 5.2. Comparison with Existing Approaches

The proposed framework for selecting security measures in a Smart City environment was compared with several existing approaches that are commonly used in the field of cybersecurity. The aim of this comparison is to show how our approach differs in terms of methodology, flexibility, and responsiveness to the specific needs of urban data systems. The most commonly used approaches in practice include static security catalogs, such as ISO/IEC 27001 Annex A or NIST SP 800-53B [2]. These frameworks provide general lists of security controls that are universally applicable but often lack context sensitivity. For example, they do not consider the operational criticality of individual services or the impacts of disruptions on the continuity of urban processes.

Another important approach option is ontological models, such as Unified Cyber Ontology [26] (UCO) or CASE, which allow for a formal description of security entities, relationships, and events [28]. These models are suitable for standardization and interoperability but often lack a direct link to operational impacts and are not designed to prioritize measures according to business or public services.

It is also necessary to consider the possibilities of using AI for anomalous detection, which focuses on identifying deviations in network traffic or system behavior [21]. This approach is very useful for detecting attacks in real time, but does not address the selection of preventive measures or work with impact metrics, such as RTO or MTDL.

Unlike the above approaches, our framework integrates Business Impact Analysis (BIA) metrics—specifically Recovery Time Objective (RTO), Maximum Tolerable Period of Disruption (MTPD), and Maximum Tolerable Data Loss (MTDL)—directly into process models using BPMN-SC extensions (developed as part of the ARTISEC project) [25]. This ensures a direct link between security measures and operational requirements of individual services. It also considers the operational criticality of data flows, allowing for prioritization of measures according to the impact on service continuity.

For example, measures that protect data with high confidentiality and low outage tolerance are prioritized over those that protect less critical processes. Importantly, the proposal uses the algorithmic selection of measures, specifically the Greedy algorithm and the genetic algorithm, which optimize the selection of measures with regard to risk coverage and efficiency. This eliminates subjectivity and ensures a transparent and repeatable selection process. Last but not least, our solution supports simulation and adaptation: thanks to the use of algorithms, various scenarios can be easily tested, for example, when the budget changes, the criticality of services changes, or when new threats are introduced.

This algorithmic approach also enhances the auditability of decisions, which is crucial for transparency and regulatory compliance in public administration.

Empirical results show that this approach leads to a higher level of risk coverage, better alignment of measures with operational needs, and higher transparency of the decision-making process. For example, in the case study of the Hradec Kralove Region, algorithmic selection led to the identification of measures that could be overlooked during manual selection, because their benefit was only apparent in combination with other measures.

While traditional approaches often rely on expert judgment or static templates, our framework enables dynamic, data-driven decision-making that is tailored to the specific urban environment and its services. This is especially important in the context of Smart Cities, where not only technology but also citizen expectations and regulatory requirements are changing [16,18]. In conclusion, the proposed framework does not represent a replacement for existing standards, but a complementary and expanding layer that enables their effective application in the context of urban services. By combining business process modeling, impact metrics, and algorithmic selection of measures, our approach provides a practical tool for managing cyber risks in an open data and IoT environment.

## 6. Conclusions

This study responded to the growing demand for systematic and measurable cybersecurity planning in Smart City environments, with a focus on the often-overlooked domain of open data. By integrating Business Impact Analysis (BIA) with process modeling via BPMN-SC, the research introduced a replicable framework that allows municipalities to quantify disruption tolerances and align security measures with service continuity objectives.

The proposed methodology was systematically developed across six sections. Section 2 synthesized existing literature and standards, identifying key limitations in static baseline approaches. Section 3 defined impact criteria and formalized the BIA annotation of Smart City processes. Section 4 presented an algorithm for the optimized selection of security controls based on risk coverage and measure efficiency. Section 5 empirically validated this algorithmic approach using simulated data for 40 measures and 14 risks, comparing a Greedy heuristic and a Genetic algorithm to assess trade-offs between speed and optimality.

As part of the ARTISEC project, the methodology was applied to the Open Data Platform of the Hradec Kralove Region. Key publication processes were modeled using BPMN-SC, enriched with RTO, MTPD, and MTDL metrics. These enriched diagrams enabled the automated mapping of cyber risks to relevant mitigation controls drawn from a structured baseline catalog.

This implementation provides tangible proof that cybersecurity planning can evolve beyond compliance to become data-driven, transparent, and aligned with the real operational demands of Smart City systems. The approach supports NIS2 Directive implementation and advances ISO/IEC 27001-aligned continuity strategies.

Future research should focus on expanding the automation potential of this framework, particularly by incorporating AI-driven adjustments to BIA parameters based on live telemetry and threat intelligence. Additional work is also warranted in extending the baseline catalog to other public sector domains, such as healthcare, transportation, or utilities, and in modeling interdependencies among controls.

Future research directions include expanding the catalog of measures to include domain-specific scenarios (e.g., healthcare, transportation), integrating with automated audit tools, and using AI to adaptively adjust BIA parameters based on telemetry data. It is also worth exploring the possibilities of connecting with digital twins of urban processes to simulate the impacts of cyber incidents.

Overall, this study lays a solid foundation for municipalities and infrastructure operators to strengthen cybersecurity governance through process-integrated, impact-sensitive, and algorithmically supported decision-making.

## Figures and Tables

**Figure 1 sensors-25-04966-f001:**
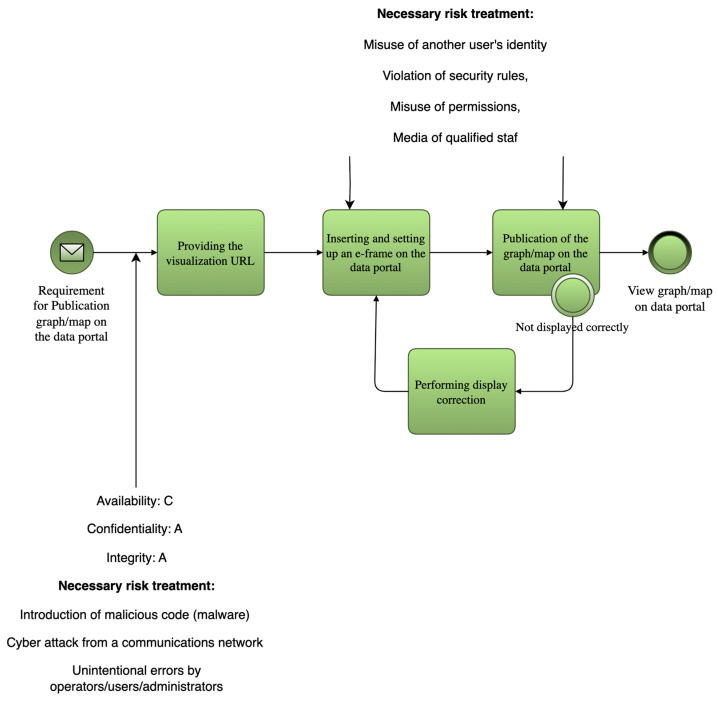
BPMN-SC model: the following essay sets out the process of publishing visualizations on the data portal.

**Figure 2 sensors-25-04966-f002:**
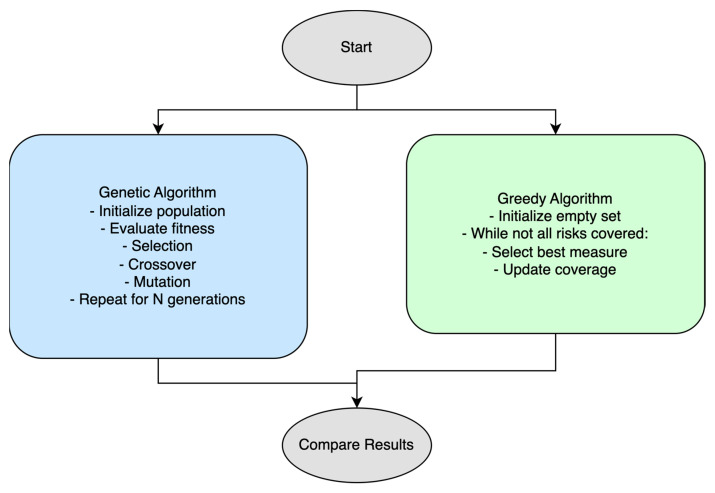
Graphic interpretation of algorithms.

**Table 1 sensors-25-04966-t001:** Comparative analysis.

Approach	Process Modeling Coverage	Quantitative Risk Metrics Support	Smart City Functional Adaptability
BPMN-SC (our approach)	Full process mapping including actors and data flow	RTO, MTPD, and MTDL embedded in diagrams	Domain-specific workflows tailored to Smart City
ISO 27001/NIST baseline	Basic process control documentation	No direct impact quantification	General-purpose standards, not city-specific
Ontology-based methods	Hierarchical structure of concepts and interactions	Limited integration with measurable indicators	Context-aware but lacking process granularity
AI-based anomaly detection	No process modeling; focuses on pattern detection	Dynamic anomaly scoring based on system inputs	Adaptable to smart infrastructure telemetry

**Table 2 sensors-25-04966-t002:** Impact criteria.

Impact Level	Financial Impact	Reputational Impact	Personal Safety Impact
N/A	Not filled in or no impact.	Not filled in or no impact.	Not filled in or no impact.
Low	Financial or material losses up to CZK 5M	Major disruption of essential services or daily life for up to 250 people.	Up to 10 people injured, requiring hospitalization for more than 24 h.
Medium	Losses up to CZK 50M	Major disruption for up to 2500 people.	Up to 10 deaths or up to 100 injured, requiring hospitalization over 24 h.
High	Losses up to CZK 500M	Major disruption for up to 25,000 people.	Up to 100 deaths or up to 1000 injured, requiring hospitalization over 24 h.
Critical	Losses exceeding CZK 500M	Major disruption for more than 25,000 people.	Over 100 deaths and more than 1000 injured, requiring hospitalization over 24 h.

**Table 3 sensors-25-04966-t003:** Defining RTO and RPO parameters.

Classification	RTO Example	RPO Example	Description
Tier 1	0–15 min	≤5 min	Life-critical systems (e.g., emergency data)
Tier 2	≤2 h	≤15 min	Core services (e.g., identity management)
Tier 3	≤24 h	≤1 h	Administrative data, non-real-time logs
Tier 4	≤72 h	≤24 h	Public archives, open data repositories

**Table 4 sensors-25-04966-t004:** BIA evaluation summary.

Attribute	Score	Level	Description	Protection Strategy
Availability	C	Low	Disruption is tolerable, and a longer recovery time (up to one week) is acceptable.	Regular data backups are sufficient to protect availability.
Confidentiality	A	High	Data must not be publicly accessible prior to publication and is subject to legal or contractual protection.	Access must be logged and controlled; external communications protected by cryptographic means.
Integrity	A	High	Integrity breaches can significantly harm business interests.	Track all changes and responsible users and enforce cryptographic protection for data transfers across external networks.
MIPD	BE	-	No moderate impact detected.	No RTO defined for recovery of single component failures.
MTPD	BE	-	No high impact detected.	No RTO defined for disaster recovery.
MTDL	24H	-	Backup must be performed at least once daily.	Daily backup schedule must be enforced.

**Table 5 sensors-25-04966-t005:** Risk analysis results.

Risk Source	Risk Level	Commentary
Identity theft	High	Systematic measures must be initiated; long-term unacceptable risk.
Security breaches, misuse of permissions or media	High	Systematic measures must be initiated; long-term unacceptable risk.
Cyber-attack via network	Medium	Risk can be reduced with cost-effective measures or may be acceptable despite higher costs.
Introduction of malware	High	Systematic measures must be initiated; long-term unacceptable risk.
Hardware/software failure	Low	Risk considered acceptable.
Support service outage (e.g., power, cooling)	Low	Risk considered acceptable.
Communication network outage	Low	Risk considered acceptable.
Unintentional operator/user/administrator errors	High	Systematic measures must be initiated; long-term unacceptable risk.
Natural disasters	Low	Risk considered acceptable.
Lack of qualified personnel	High	Systematic measures must be initiated; long-term unacceptable risk.
Physical security breach (theft, vandalism, sabotage)	Low	Risk considered acceptable.

**Table 6 sensors-25-04966-t006:** Mapping of risks to recommended controls.

Risk Description	Recommended Measures from the Baseline Catalog
Misuse of another user’s identity	O3, O6, O21, O37, O40, O43
Violation of security rules, misuse of permissions/media	O2, O3, O6, O7, O8, O9, O11, O13, O14, O17, O18, O21, O22, O27, O28, O29, O30, O31, O32, O33, O37, O40, O41, O42, O43, O44, O49
Cyber-attack from communication networks	O2, O3, O4, O5, O6, O7, O8, O9, O16, O17, O18, O19, O20, O21, O22, O27, O37, O38, O39, O40, O43
Introduction of malware	O10, O16, O43, O48
Unintentional errors by operators/users/administrators	O7, O8, O9, O28, O29, O30, O32, O33, O41, O42, O49
Lack of qualified personnel	O27, O28, O29, O30, O31, O32

## Data Availability

All the data could be found in the references.

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
