# Peer review of "Cybersecurity Baseline and Risk Mitigation for Open Data in IoT-Enabled Smart City Systems: A Case Study of the Hradec Kralove Region"

_sensors, 2025, doi:10.3390/s25164966_

Round 1
Reviewer 1 Report
Comments and Suggestions for Authors
Write out the full meaning BPMN in this abstract
It will be more helpful and interesting if the results of the impact of disruptions in terms of Recovery Time Objective (RTO), Maximum Tolerable Period of Disruption (MTPD), and Maximum Tolerable 17 Data Loss (MTDL) can be reported in the abstract.
Is it your journal standard or format to use Chapters instead of sections? Chapter is commonly used in book chapter not in an article. Page 2
If RTO = 0, this implies fully redundant infrastructure but what of if RTO=1 page 6
I can’t see the results findings based on the impact of disruptions in terms of Recovery Time Objective (RTO), Maximum Tolerable Period of Disruption (MTPD), and Maximum Tolerable 17 Data Loss (MTDL) in the results discussion section.
Author Response
Dear reviewer,
thank you very much for your valuable comments, which will certainly help to improve the quality of the article itself.
Comment1: It will be more helpful and interesting if the results of the impact of disruptions in terms of Recovery Time Objective (RTO), Maximum Tolerable Period of Disruption (MTPD), and Maximum Tolerable 17 Data Loss (MTDL) can be reported in the abstract.
Response1: AGREE. We have expanded the abstract of the article with an example of MTDL from the Open Data domain. Due to the limitations of the abstract, further expansion is integrated directly into the chapters of the article.
Comment2: Is it your journal standard or format to use Chapters instead of sections? Chapter is commonly used in book chapter not in an article. Page 2
Response2: AGREE. Chapters were replaced by sections, according to the journal template.
Comment3: If RTO = 0, this implies fully redundant infrastructure but what of if RTO=1 page 6
Response3: Agree. We added an explanation for RTO 1 in the text of the relevant section.
Comment4: I can’t see the results findings based on the impact of disruptions in terms of Recovery Time Objective (RTO), Maximum Tolerable Period of Disruption (MTPD), and Maximum Tolerable 17 Data Loss (MTDL) in the results discussion section.
Response4: Agree. The Discussion and results section has been significantly revised and expanded. Including links to BIA, MTDL and RTO in several parts of this section. Details on genetic and greedy algorithms have also been added, including their comparison and definition against other standards and approaches used in this area.
We have also revised other sections in response to other reviewer comments and adjusted the numbering of citations to add new ones.

Reviewer 2 Report
Comments and Suggestions for Authors
Recommended Changes:
- Add related works about cybersecurity risks and vulnerabilities associated with open data in IoT-enabled smart city infrastructures, and explain how these risks impact data confidentiality, integrity, and availability. Include more up-to-date references.
- Compare your solution against existing works in an evaluation section. A better clarification on this is needed.
- Instead of copying the source code, present it using algorithm presentations, which is better for readers (see pages 16-18).
Improve the discussion and results sections to reflect a deeper explanation of the type of work pursued, to help readers properly understand the results. - Clarify the difference between the implementation of the Greedy algorithm and the implementation of the genetic algorithm. It is unclear what the implications of these algorithms are.
- Authors should investigate more critical related works and add those works to the manuscript. Also, authors need to describe future research issues, challenges, and works for potential researchers.
- Figure 1 is not explained. More details must be discussed and justified. The quality of the figure needs improvement to make it more readable.
Author Response
Dear reviewer,
thank you very much for your valuable comments, which will certainly help to improve the quality of the article itself.
Comment1: Add related works about cybersecurity risks and vulnerabilities associated with open data in IoT-enabled smart city infrastructures, and explain how these risks impact data confidentiality, integrity, and availability. Include more up-to-date references.
Response1: AGREE. The related work section focusing on Open Data and related areas has been added.
Comment2: Compare your solution against existing works in an evaluation section. A better clarification on this is needed.
Response2: AGREE. The Discussion and results section has been significantly revised and expanded. Including links to BIA, MTDL and RTO in several parts of this section.
Comment3: Instead of copying the source code, present it using algorithm presentations, which is better for readers (see pages 16-18).
Response3: AGREE. Algorithm presentations focusing on genetic and greedy algorithms have been added.
Comment4: Improve the discussion and results sections to reflect a deeper explanation of the type of work pursued, to help readers properly understand the results.
Response4: AGREE. As part of the overall revision, the discussion and results sections were added to better explain the comparison of existing approaches and to define the benefits of the proposed solution for better understanding.
Comment5: Clarify the difference between the implementation of the Greedy algorithm and the implementation of the genetic algorithm. It is unclear what the implications of these algorithms are.
Response5: AGREE. It is newly presented in a separate subsection 5.1 Comparison of Greedy and Genetic Algorithms for Risk Mitigation
Comment6: Authors should investigate more critical related works and add those works to the manuscript. Also, authors need to describe future research issues, challenges, and works for potential researchers.
Response6: AGREE. The related work and discussion and result sections have been revised and supplemented, including relevant publications and renumbering of citations.
Comment7: Figure 1 is not explained. More details must be discussed and justified. The quality of the figure needs improvement to make it more readable.
Response7: AGREE. The description of Figure 1 has been added.
We have also revised other sections in response to other reviewer comments.

Round 2
Reviewer 2 Report
Comments and Suggestions for Authors
As outlined in the previous review, the manuscript requires substantial revisions to be considered for publication.
Comment 3: Instead of copying the source code, please present it using algorithm presentations, which is better for readers (see pages 16-18). The authors have not addressed the first reviewer's request and have only copied the source code. Instead of copying and pasting the source code, please present an algorithm presentation for each Python implementation.
Comment 2: Compare your solution against existing works in an evaluation section. A better clarification on this is needed. The request has been addressed but without any supporting references.
Comment 7: Figure 1 is not explained, and its relation to the presented algorithm is unclear. Additionally, the authors have included a reviewer's comment: "For better readability, we recommend increasing the contrast and text size in the diagram." Furthermore, the accompanying image is blurry. The same issues apply to Figure 2.
The other requests have been addressed.
Author Response
Dear reviewer,
thank you very much for your valuable comments, which will certainly help to improve the quality of the article itself. The changes have been highlighted with green color.
Comment 3: Instead of copying the source code, please present it using algorithm presentations, which is better for readers (see pages 16-18). The authors have not addressed the first reviewer's request and have only copied the source code. Instead of copying and pasting the source code, please present an algorithm presentation for each Python implementation.
Response3: AGREE. Individual algorithms and their comparison were supplemented with a presentation of the Python implementation.
Comment2: Compare your solution against existing works in an evaluation section. A better clarification on this is needed. The request has been addressed but without any supporting references.
Response2: AGREE. The Discussion and Results section has been revised and expanded with supporting references.
Comment 7: Figure 1 is not explained, and its relation to the presented algorithm is unclear. Additionally, the authors have included a reviewer's comment: "For better readability, we recommend increasing the contrast and text size in the diagram." Furthermore, the accompanying image is blurry. The same issues apply to Figure 2.
Response7: AGREE. Additional explanation in the form of an example and image description has been added, and the image quality has been improved.
